# Phosphorus Metabolism and Function in Ruminants: Current Knowledge

**DOI:** 10.3390/ani16010130

**Published:** 2026-01-02

**Authors:** Beata Abramowicz, Ewa Tomaszewska, Oliwia Brzezińska, Karolina Kłos, Miroslav Urosevic, Łukasz Kurek

**Affiliations:** 1Department and Clinic of Animal Internal Diseases, University of Life Sciences in Lublin, Akademicka 13, 20-912 Lublin, Poland; beata.abramowicz@up.edu.pl (B.A.);; 2Department of Animal Physiology, Faculty of Veterinary Medicine, University of Life Sciences in Lublin, Akademicka 13, 20-912 Lublin, Poland; ewa.tomaszewska@up.lublin.pl; 3Faculty of Veterinary Medicine, University of Life Sciences in Lublin, Akademicka 13, 20-912 Lublin, Poland; 4Department of Animal Science, Faculty of Agriculture, University of Novi Sad, Trg D. Obradovića 8, 21000 Novi Sad, Serbia; miroslav.urosevic@stocarstvo.edu.rs

**Keywords:** phosphorus deficiency, dairy cows, mineral metabolism, FGF23, acute-phase proteins, postpartum disorders

## Abstract

Phosphorus is an essential mineral that supports bone formation, energy metabolism, and normal muscle and nerve function in dairy cattle. Around calving, cows often experience a negative energy and mineral balance, which may lead to a drop in blood phosphorus levels, known as hypophosphatemia. This condition can cause weakness, loss of appetite, anemia, and decreased milk production. If left untreated, it may progress to serious disorders such as postpartum hemoglobinuria or downer cow syndrome. This review explains how phosphorus metabolism is regulated in the body, especially by hormones such as parathyroid hormone, vitamin D, and Fibroblast Growth Factor 23 (FGF23). It also highlights how phosphorus deficiency affects red blood cells and immune function, and discusses practical ways to monitor and prevent it through proper feeding strategies. Understanding and managing phosphorus balance is crucial for maintaining the health, welfare, and productivity of dairy cows.

## 1. Introduction

Phosphorus (P) is an essential macromineral required for normal function of the animal organism. It plays a pivotal role in numerous physiological processes, including energy metabolism, nerve conduction, regulation of acid–base balance, and mineralization of bones and teeth. Approximately 80–85% of total body phosphorus resides in the skeleton. The remainder is distributed within body fluids and cells, where it serves structural (phospholipids), energetic ATP (adenosine triphosphate) ADP (adenosine diphosphate) AMP (adenosine monophosphate), and regulatory functions (protein phosphorylation, intracellular signaling molecules).

There are substantive interspecies differences in phosphorus metabolism, particularly between ruminants and monogastrics. In ruminants, the rumen microbiota substantially modifies phosphorus availability and the route and efficiency of its absorption. In contrast, in monogastric species, phosphorus supplied in organic form, especially as phytate, may be poorly available in the absence of enzymes such as phytase.

Phosphorus deficiency has significant implications for dairy productivity and animal welfare, particularly in regions with naturally P-depleted soils. Chronic inadequacy of dietary phosphorus reduces milk yield, prolongs calving intervals, increases the incidence of periparturient disorders, and impairs skeletal development in replacement heifers, resulting in substantial economic losses at the herd level.

The aim of this review is to provide a comprehensive synthesis of current knowledge on the physiological role of phosphorus in animals, its metabolism, and its clinical relevance, with particular emphasis on cattle. It further examines the practical implications of phosphorus deficiency for the underlying mechanisms, clinical course, diagnosis, and treatment of phosphorus-deficiency-related diseases in cattle.

## 2. Physiology of Phosphorus

### 2.1. Chemical Forms and Bioavailability

Phosphorus in animals exists as inorganic phosphate (Pi) and in multiple organic forms including ATP, phospholipids, and nucleic acids. Approximately 80–85% of total body phosphorus resides in bone as hydroxyapatite, whereas the remaining fraction is present intra- and extracellularly and participates in essential cellular processes [1].

Bioavailability depends on chemical form and gastrointestinal pH. Inorganic phosphates are generally well soluble and efficiently absorbed. In plant-derived feeds, phosphorus predominantly occurs as phytate, which is poorly available to monogastrics due to the lack of endogenous phytase [2]. In ruminants, rumen microorganisms express phytases that hydrolyze phytate and release absorbable phosphate [2]. Abomasal acidity enhances phosphate dissolution, whereas salivary phosphates buffer rumen contents and remain accessible to the microbiota at physiological rumen pH.

### 2.2. Gastrointestinal Absorption (Ruminants vs. Monogastrics)

Phosphate absorption occurs primarily in the duodenum and jejunum. In monogastrics, uptake depends on solubility and epithelial transporters; phytate-bound phosphorus is poorly available (<30%) without exogenous phytase [2].

Ruminants efficiently utilize phosphorus because the rumen microbiota hydrolyzes phytate, making phosphorus source less critical. Furthermore, extensive endogenous recycling occurs: large amounts of Pi are secreted into saliva, enter the rumen, support microbial metabolism, and are then reabsorbed in the intestine. Apparent absorption in cattle is typically 60–70%, compared with <30% in monogastrics without phytase supplementation [2,3].

### 2.3. Endocrine Regulation (PTH, Calcitriol, FGF23)

Phosphate homeostasis is regulated by parathyroid hormone (PTH), calcitonin, calcitriol, and fibroblast growth factor 23 (FGF23), with minor contributions from other hormones [2,4,5]. PTH mobilizes Ca and P from bone, stimulates renal calcitriol synthesis, and decreases renal phosphate reabsorption, increasing phosphaturia [2,4,5]. In ruminants, it also stimulates salivary phosphate secretion to the rumen [2]. Calcitriol enhances intestinal absorption of Ca and P, and its synthesis is stimulated by PTH and inhibited by FGF23 [4,5]. FGF23 decreases renal phosphate reabsorption and suppresses calcitriol formation [4,5]. These coordinated actions maintain plasma phosphate within narrow physiological limits. In addition to its regulatory role, FGF23 is now evaluated as a potential biomarker of phosphate stress, with early postpartum spikes shown to precede measurable hypophosphatemia in high-yielding dairy cows.

### 2.4. Renal Regulation and Recycling

Virtually all plasma Pi is freely filtered by the glomerulus, and >99% is normally reabsorbed in the proximal tubule in healthy ruminants [3,4]. In monogastrics and young calves, renal excretion plays a greater role in maintaining phosphate balance [3,4].

Phosphate excretion increases during metabolic acidosis and with elevated PTH. Acidosis reduces tubular reabsorption to provide titratable buffers for urinary acid excretion, while PTH directly inhibits reabsorption and promotes phosphaturia [2,4,5].

In severe hypophosphatemia, renal conservation mechanisms can suppress urinary phosphate loss almost completely [2,4,5]. These dynamics are relevant in cows fed high-concentrate diets that predispose to ruminal acidosis and in dry cows fed anionic diets, both of which increase urinary phosphate elimination [2,4].

### 2.5. Distribution: Bone, Extracellular, and Intracellular Compartments

Bone constitutes approximately 80–85% of total body phosphorus and serves as the primary dynamic reservoir. During periods of heightened metabolic demand—particularly in early lactation—skeletal phosphorus is mobilized to maintain plasma inorganic phosphate and support milk synthesis, followed by gradual remineralization once dietary intake improves [2,4,6].

The remaining phosphorus is distributed between the extracellular and intracellular compartments. Although the extracellular pool is small, plasma phosphate is tightly regulated due to its essential role in ATP production, nucleic acid synthesis, and enzyme activity. Intracellular phosphorus supports high turnover metabolic pathways and accounts for the majority of non-skeletal phosphorus [2,4,6].

Phosphate exchange between compartments is highly dynamic. Insulin promotes rapid cellular uptake of phosphate, causing a transient decline in plasma Pi, a response commonly observed following hyperinsulinemia or intravenous glucose administration. Similarly, catecholamines can enhance cellular phosphate transport and stimulate parathyroid hormone release, contributing to stress-associated hypophosphatemia. These redistributions may produce marked short-term fluctuations in serum Pi without reflecting true phosphorus depletion [6,7,8,9].

Despite such trans compartmental shifts, homeostasis is maintained through coordinated skeletal buffering, renal excretion, and endocrine regulation. Consequently, interpretation of circulating phosphate must consider the animal’s metabolic and physiologic state to avoid misclassification of adaptive, transient changes as evidence of true deficiency [6,7,8,9].

## 3. Functional Roles of Phosphorus at the Cellular Level

### 3.1. Skeletal System. Effects on Growth, Bone Mineral Density, and Dentition—Pathologies: Osteodystrophy, Skeletal Deformities, Rickets, and Osteomalacia

Phosphorus, together with calcium, is a principal mineral constituent of bones and teeth. It forms hydroxyapatite crystals (Ca_10_(PO_4_)6(OH)_2_) deposited within the collagenous matrix, which confer bone hardness and strength. Proper mineralization of bone and teeth depends on adequate supplies of both phosphorus and calcium, as well as vitamin D. Phosphorus influences skeletal growth and bone mineral density. In growing animals, deficiency disrupts endochondral ossification at the growth plates, causing rickets; in adults, it results in inadequate mineralization of previously formed bone (osteomalacia). Rickets manifests clinically with skeletal deformities (e.g., limb bowing, metaphyseal enlargement), whereas osteomalacia in dairy cows may present as bone pain, locomotor difficulty, and a propensity for rib or pelvic fractures. Moreover, prolonged phosphorus deficiency in the face of relatively high dietary calcium favors secondary hyperparathyroidism and fibrous osteodystrophy: bone becomes demineralized and is replaced by fibrous tissue (notably in craniofacial and long bones), a pattern described in young cattle on diets with severely imbalanced Ca:P ratios (and more commonly reported in other domestic species such as horses and goats). Clinically, phosphorus deficiency in growing stock presents with stunted growth and fracture susceptibility; in adults, with musculoskeletal pain, difficulty rising, and pica (abnormal mineral appetite, chewing bones, wood) [2,10,11,12]. These pathologies (osteodystrophy, rickets, osteomalacia) underscore the critical role of phosphorus in maintaining the structural integrity of the skeletal system.

### 3.2. Phosphorus in Muscular Function, Energy Metabolism, and Membrane Integrity

Phosphorus plays an essential role in cellular energetics and membrane structure, making it fundamental to normal muscle function. ATP and phosphocreatine serve as the primary high-energy phosphate carriers in muscle, enabling rapid energy transfer for actin–myosin cross-bridge cycling and sustaining contractile activity. Numerous intermediates of glycolysis and the Krebs cycle require phosphorylation, so efficient ATP regeneration depends on adequate inorganic phosphate availability [2].

Phosphate deficiency disrupts these ATP-dependent processes, leading to reduced muscle strength, impaired contractility, and accelerated fatigue. In dairy cattle, profound hypophosphatemia may manifest as postpartum recumbency, a downer cow syndrome distinct from hypocalcemia. Affected cows remain unable to rise despite normal calcium concentrations; in many such cases, phosphorus depletion limits ATP generation in skeletal muscle. Approximately one-third of cows failing to respond to calcium therapy for milk fever are concurrently severely hypophosphatemic, and intravenous phosphate administration typically restores the ability to stand [13].

Beyond skeletal muscle, phosphorus deficiency also compromises myocardial contractility. Although overt cardiac signs are uncommon in cattle, evidence from other species indicates that severe hypophosphatemia can depress cardiac performance and predispose to arrhythmias.

Phosphorus is further required for the synthesis of 2,3-diphosphoglycerate (2,3-DPG) in erythrocytes, influencing hemoglobin’s oxygen-binding affinity. Phosphate deprivation lowers erythrocyte ATP and disrupts Na^+^/K^+^-ATPase activity, impairing membrane fluidity and ion balance. These abnormalities can lead to hemolysis and are implicated in postparturient hemoglobinuria. Collectively, inadequate phosphorus intake destabilizes cell membranes, diminishes oxygen delivery capacity, and impairs ATP-dependent functions across muscle and blood cells [14,15,16].

### 3.3. Phosphorus and Rumen Microbial Metabolism

In the rumen, available phosphate is crucial for microbial growth, fermentation, and fiber digestion. Rumen microbes incorporate phosphorus into nucleic acids, phospholipids, and cofactors, so they require a steady supply of inorganic phosphate. Cellulolytic bacteria in particular have high P requirements (higher than amylolytic bacteria) [1], so low dietary P impairs fiber fermentation. Ruminants recycle large amounts of phosphate via saliva: about 3–4 times the daily intake of P is secreted in saliva to buffer rumen acid and supply microbes. Salivary P not only buffers pH but also maintains soluble phosphate levels that support microbial metabolism. Ruminal bacteria also produce phytase enzymes to free phosphate from plant phytate, but efficiency of phytate degradation depends on diet (e.g., grain processing) and retention time.

If P is deficient, rumen fermentation deteriorates (Table 1). Low dietary P reduces salivary P output and ruminal soluble phosphate, leading to slower microbial growth. Studies report that P deficiency raises rumen fluid pH and ammonia (NH3–N) concentrations while suppressing total volatile fatty acid (VFA) production and microbial protein yield. Crucially, fiber digestion drops markedly: in one survey of rumen cultures, cellulose and hemicellulose breakdown were significantly reduced under P-depletion. Starch fermentation was relatively preserved, reflecting the lower P needs of amylolytic bacteria. As a result, phosphorus-deficient rumens have more residual fiber and less microbial biomass.

Adequate phosphorus nutrition (often via saliva) is therefore critical for normal rumen function. Without it, cows lose feeding drive and milk production as rumen microbes cannot fully degrade forages or synthesize enough microbial protein [1].

### 3.4. Role in pH Buffering—Phosphate Buffers (Particularly Important in Body Fluids): Importance for Acid–Base Balance

The phosphate buffer system is one of the major mechanisms maintaining systemic acid–base balance. In body fluids, particularly intracellular fluid and plasma, it comprises two principal components: the monohydrogen phosphate ion (HPO_4_^2−^), acting as a weak base, and the dihydrogen phosphate ion (H_2_PO_4_^−^), its conjugate weak acid. The pK_a of the H_2_PO_4_^−^/HPO_4_^2−^ pair is approximately 7.2; thus, at physiologic blood pH (~7.4) both forms are present in comparable proportions, which is optimal for buffering. When a strong acid (e.g., HCl) is introduced, the basic HPO_4_^2−^ ion binds protons, converting to H_2_PO_4_^−^ and thereby neutralizing most of the added H^+^ [11,17]. Analogously, in the presence of a strong base (e.g., NaOH), the dihydrogen phosphate ion donates a proton and converts to HPO_4_^2−^, thereby buffering excess OH^−^. The reactions proceed as follows:HPO_4_^2−^ + H^+^ → H_2_PO_4_^−^ (proton binding by phosphate—acid neutralization)(1)H_2_PO_4_^−^ + OH^−^ → HPO_4_^2−^ + H_2_O (proton donation—base neutralization)(2)

These equilibria blunt large swings in pH, protecting cells from injury due to acidosis or alkalosis. The phosphate buffer accounts for ~5–10% of whole-blood buffering capacity (the carbonate system is dominant) but plays a more prominent role intracellularly, where phosphate concentrations are higher, and in urine, where it is central to titratable acid excretion [12].

Phosphate is particularly important in the kidney: during urinary acidification, phosphate ions enable excretion of excess H^+^ as acidic salts. Renal tubules secrete protons that bind to filtered HPO_4_^2−^, forming H_2_PO_4_^−^, which is excreted in the urine as titratable acidity. This mechanism removes metabolically generated hydrogen ions from the body, preventing excessive acidemia. In ruminants, salivary phosphates also provide substantial buffering in the rumen, neutralizing a portion of the acids produced by fermentation (e.g., volatile fatty acids) and protecting against declines in rumen pH. Due to these phosphate pools, the organism maintains stable pH in body fluids despite continual acid production, which is critical for enzymatic function and biochemical processes [12].

## 4. Clinical and Pathophysiological Consequences of Phosphorus Deficiency in Cattle

### 4.1. Neuromuscular Effects: Muscle Weakness, Recumbency, “Alert Downer Cow”

In hypophosphatemia, myocytes (muscle cells) lack sufficient phosphate to generate ATP and phosphocreatine. This energy failure manifests as profound muscle weakness. Affected cows may walk stiffly or collapse due to muscle fatigue. Laboratory and field reports note muscle pain and soreness in phosphate-deficient cows. Periparturient hypophosphatemia is associated with anorexia, muscle weakness, muscle or bone pain, and even rhabdomyolysis (muscle breakdown). Weak ATP-driven ion pumps also impair muscle membrane potentials and contractility. Indeed, hypophosphatemic cattle often have reduced contraction strength in skeletal and cardiac muscle, as ATP stores dwindle.

Clinically, hypophosphatemic cows that collapse often remain bright and alert despite being recumbent. These “alert downer” cows show little systemic illness. By definition, an alert downer is involuntarily recumbent but has a normal appetite, mentation, and vital signs. In practice, alert-downer cows should prompt evaluation of electrolyte and mineral status. The most common cause of periparturient recumbency is hypocalcemia (milk fever), in which cows are depressed and flaccid and promptly improve with IV calcium. In contrast, truly phosphorus-deficient cows usually do not respond to calcium infusion. If hypophosphatemia is present (often in early lactation), calcium-treated cows may remain down. Intravenous or oral phosphate supplementation (e.g., potassium phosphate) typically restores muscle function in these cases. However, controlled studies have been unable to reproduce a definitive P-depletion downer syndrome, suggesting hypophosphatemia is at most an occasional contributor rather than a sole cause of recumbency.

### 4.2. Immune System Dysfunction and Increased Disease Susceptibility

Phosphorus is also essential for immune cell metabolism. Recent studies [12,18] highlight considerable variability in the magnitude of immune suppression associated with moderate hypophosphatemia, suggesting that inflammatory status, parity, and concurrent trace-mineral deficiencies may modulate the response. Phosphate availability directly affects lymphocyte proliferation through its role in nucleotide synthesis, mitochondrial ATP generation, and mTOR-dependent cell-cycle progression. In macrophages, Pi deficiency downregulates MAPK and NF-κB signaling, resulting in reduced cytokine release (IL-1β, IL-6) and impaired priming of innate responses. Neutrophils exhibit decreased oxidative burst capacity due to reduced NADPH generation and impaired membrane phosphorylation dynamics. Leukocytes depend on rapid ATP generation to power phagocytosis, motility, and cytokine production. In phosphorus deficiency, neutrophils and other leukocytes cannot produce optimal energy. Reduced intracellular ATP impairs multiple immune functions. Experimental P depletion in dairy cows has been shown to decrease circulating granulocyte counts and reduce their survival after phagocytosis [19]. Although lymphocyte proliferation was not significantly affected, the decline in neutrophil function indicates a clear innate immunosuppression.

In practice, phosphorus-deficient cattle exhibit higher rates of infection in early lactation. Suboptimal neutrophil and macrophage activity leaves cows prone to uterine infections and mastitis. Indeed, farms with low P often report more metritis and mastitis cases. Phosphorus deficiency may also blunt vaccine responses, presumably by limiting cytokine signals needed for antibody production. Overall, chronic or periparturient hypophosphatemia compromises immune defense, predisposing cows to a variety of infectious diseases. Timely correction of P deficits (by feeding or supplementation) can help restore normal leukocyte function and disease resistance [19]. Subclinical hypophosphatemia, even without overt clinical signs, may progressively weaken host defenses by reducing neutrophil recruitment efficiency, impairing macrophage phagocytosis, and prolonging uterine involution postpartum. Herd-level studies indicate that cows with persistently low serum Pi have increased risks of metritis, subclinical mastitis, and poorer vaccine responses.

### 4.3. Transcompartmental Shifts in Phosphate Distribution

Serum phosphate levels can change rapidly due to cellular shifts, independent of true body P stores. Insulin and stress hormones drive phosphate into cells during anabolic metabolism. For example, oral or IV glucose and parenteral insulin administration cause a brisk uptake of phosphate by muscle and liver, precipitating marked hypophosphatemia even though total P is unchanged. Respiratory or metabolic alkalosis (from hyperventilation or rumen bloat) similarly enhance cellular phosphate entry. Likewise, acute stress and catecholamine release (e.g., epinephrine surge) activate glycolysis and Na^+^/K^+^ pumps, pulling phosphate into cells. These effects can make serum phosphate transiently low during recovery from milk fever or after oral drenching with dextrose. Because of this, a single blood phosphate measurement can mislead: normal cattle may appear hypophosphatemic in response to insulin or alkalemia, and conversely plasma phosphate may rise if cows catabolize muscle (releasing P) or mobilize bone.

In clinical practice, it is important to interpret serum phosphate in context. Apparent hypophosphatemia in a healthy fresh cow (e.g., during the first days of lactation) may simply reflect the natural shift of P into milk or cells, and not indicate a pathological deficit. Conversely, true P deficiency should be suspected when cows show compatible signs (weakness, hemolysis, poor appetite) that are not explained by other causes, especially if they do not improve with calcium therapy. Appetite suppression is a recognized marker of long standing phosphorus depletion (Figure 1). Understanding these compartmental shifts helps avoid over-diagnosis of “hypophosphatemia” and ensures that phosphate supplementation is used appropriately in stressed or recumbent cows.

## 5. Clinical Consequences of Phosphorus Deficiency in Cattle

### 5.1. Subclinical Postpartum Hypophosphatemia

Subclinical postpartum hypophosphatemia refers to the transient decline in circulating inorganic phosphate commonly observed during the first days of lactation in high yielding dairy cows. The onset of lactation requires substantial phosphorus export into colostrum and milk. Bovine milk contains approximately 0.9 to 1.0 g of phosphorus per kilogram, which means that a cow producing 30 to 40 kg of milk daily secretes nearly 27 to 40 g of phosphorus per day through milk alone [20]. At the same time, dry matter intake is reduced in the periparturient period due to hormonal, behavioral, and metabolic changes that accompany calving [21]. This creates a temporary mismatch between phosphorus output and intake, causing a transient negative phosphorus balance with parallel mobilization of skeletal phosphorus reserves.

Herd surveys consistently demonstrate that a high proportion of fresh cows exhibit reduced plasma inorganic phosphate values during the first week postpartum, with estimates exceeding 50 percent in some populations [18]. These reductions are widely considered part of normal metabolic adaptation rather than evidence of clinical phosphorus deficiency. Alterations in mineral handling during colostrogenesis, changes in renal phosphorus excretion, and shifts in salivary phosphorus cycling are implicated in this temporary decline [22]. Provided the dietary phosphorus supply meets the nutritional recommendations and dry matter intake improves after calving, plasma phosphorus values typically normalize within seven to fourteen days.

Subclinical postpartum hypophosphatemia rarely produces overt clinical signs. Recumbency, hemoglobinuria, muscular collapse, or marked anorexia are not characteristic of this adaptive state and should not be attributed to phosphorus unless deeper deficiency or concurrent metabolic disease is documented. Some studies have speculated that mild reductions in cellular phosphorus availability may transiently modulate rumen microbial growth or early lactation immune responses. Evidence remains inconsistent, and such effects have not been definitively separated from the broader physiological stressors of fresh cow transition, including negative energy balance and oxidative demands [23].

Interpretation of serum phosphate should therefore be contextual. Acute hormonal responses, insulin mediated phosphate uptake, and redistribution between plasma and tissue compartments may render single measurements unrepresentative of total phosphorus status. In the absence of clinical signs, aggressive supplementation or corrective drenching is not warranted and may lead to unnecessary phosphorus loading. Monitoring phosphorus in the first week postpartum can inform herd level nutritional strategies, but mild declines in inorganic phosphate during this window are physiologically expected and self-resolving, provided dietary intake and rumen function stabilize appropriately [20,22,23].

### 5.2. Chronic and Clinically Significant Phosphorus Deficiency

Chronic phosphorus deficiency reflects a failure of adaptive homeostatic mechanisms and results in persistent metabolic compromise. In contrast to transient postpartum hypophosphatemia, prolonged deficiency indicates true mineral depletion, insufficient dietary replacement, and exhaustion of skeletal buffering.

Sustained phosphorus insufficiency suppresses rumen microbial proliferation, depresses cellulolytic kinetics, and reduces propionate yield. This translates not only into reduced feed efficiency but also into altered short chain fatty acid ratios that shift hepatic glucogenic flux and exacerbate negative energy balance [16,24,25,26]. Chronic hypophosphatemia has also been linked with reduced rumen epithelial turnover, impaired papillary regeneration, and dysbiosis, promoting greater ruminal endotoxin permeability into circulation. Although not a primary cause of metabolic inflammation, low phosphorus availability may amplify lipopolysaccharide exposure and subsequent inflammatory tone in long standing deficiency states.

Musculoskeletal manifestations extend beyond demineralization. Phosphorus dependent ATP flux also underpins skeletal muscle repair and basal protein turnover. Chronic depletion predisposes to subclinical myodegenerative lesions, low grade muscle fibrosis, and impaired locomotor endurance. Cows demonstrating prolonged low phosphorus availability frequently show stilted gait, reluctance to rise, and intermittent weight shifting unrelated to traumatic lameness. In grazing systems, skeletal manifestations progress to fibrous osteodystrophy, pica, and structural deformity, and advanced locomotor impairment remains consistent with chronic mineral starvation (Figure 2) [27,28]. Newer work [29] indicates that chronic low-grade P deficiency induces subtle cortical thinning detectable only through high-resolution imaging, which may precede overt clinical osteomalacia. Recent studies using DXA and micro-CT imaging demonstrate that prolonged subclinical phosphorus restriction leads to measurable trabecular rarefaction, reduced cortical thickness, and altered markers of bone turnover (P1NP, CTX-I), even before clinical osteomalacia develops.

Phosphorus deficiency may interfere with reproductive physiology through reduced ovarian ATP content, altered LH pulse frequency, impaired follicular steroidogenesis, and decreased synthesis of phospholipid-derived second messengers essential for granulosa-cell signaling. Low Pi has been associated with smaller dominant follicles, delayed ovulation, and compromised luteal function. Reproductive efficiency similarly deteriorates due to impaired uterine contractility and diminished endometrial repair capacity. Some studies report clear associations between low prepartum P and delayed ovulation, while others show minimal effect when energy balance is adequate [30], indicating a complex, multifactorial relationship. Phosphorus dependent phosphorylation pathways modulate myometrial ATP cycling, and long-term deficiency is associated with delayed uterine involution and less coordinated postpartum contractility. While retained fetal membranes are multifactorial, chronic phosphorus depletion increases susceptibility to delayed clearance processes and postpartum endometrial inflammation, especially in systems with concurrent negative energy balance.

Keratinization defects have also been reported in chronic deficiency. Phosphorus is required for epidermal lipid synthesis and structural membrane phospholipids, and long-term depletion negatively affects claw horn integrity. Cows maintained under phosphorus deficient range conditions may be predisposed to sole softening, heel erosion, and increased risk of noninfectious lameness syndromes independent of bone pain. These lesions are often overlooked but contribute significantly to locomotor compromise.

#### Periparturient Phosphorus Deficit: Failure to Respond to Calcium

Periparturient phosphorus deficiency may present clinically as recumbency that does not resolve following standard intravenous calcium administration. Such cows remain bright, maintain appetite, and continue attempts to stand, yet cannot achieve full weight bearing. In these cases, serum biochemical testing reveals a marked reduction in plasma inorganic phosphate, sometimes below 1.5 mg per deciliter [13]. Supplementation with phosphorus preparations, administered intravenously or orally, has been reported to restore muscle contractility and enable rising after treatment, confirming the role of phosphorus deficit in neuromuscular failure [13].

The pathophysiologic explanation lies in energy metabolism. Contractile function in skeletal muscle requires adequate intracellular ATP generation, which in turn depends on phosphate availability. When profound hypophosphatemia is present during the periparturient period, ATP dependent ion pumping and actomyosin cycling cannot be sustained at levels necessary for posture and limb extension. Affected cows exhibit preserved mentation and feed interest but fail mechanically to lift their own mass, differentiating this state from classic milk fever, in which recumbency is accompanied by profound flaccidity and depression.

Phosphorus depletion during the periparturient window frequently coexists with hypocalcemia because the parathyroid hormone released in response to declining calcium promotes the renal and salivary excretion of phosphate [31]. In geographic regions or feeding systems with marginal phosphorus delivery, this endocrine mechanism amplifies the likelihood of combined calcium and phosphorus insufficiency, complicating clinical interpretation. Once cows respond to phosphorus therapy and stand, corrective modification of dietary phosphorus supply remains indicated to prevent recurrence.

Accurate identification of periparturient phosphorus deficit requires measurement of plasma inorganic phosphate in addition to calcium in cases of persistent recumbency. Reliance on calcium therapy alone may mask the underlying mineral imbalance and delay recovery. Monitoring phosphorus status within the transition period therefore supports the effective management of recumbent cows and reduces the risk of prolonged neuromuscular compromise in the early stages of lactation [13,31].

## 6. Hematologic Consequences

### Hematologic Consequences of Phosphorus Deficiency and Postpartum Hemoglobinuria (PPH)

Phosphorus deficiency affects the erythron profoundly when plasma inorganic phosphate (Pi) becomes critically reduced. Acute hypophosphatemia is repeatedly associated with intravascular hemolysis and hemoglobinuria, whereas long-term deficiency commonly results in mild normocytic hypochromic anemia (Figure 3) [16,25,26,32,33]. The central mechanism involves intracellular phosphate depletion that lowers ATP and 2,3 DPG synthesis, compromises Na^+^/K^+^ ATPase function, promotes osmotic fragility, and destabilizes membrane integrity [33,34]. Organic phosphorus is essential for phospholipid turnover and structural membrane maintenance, therefore deficiency heightens erythrocyte susceptibility to rupture [35]. ATP depletion also weakens antioxidant defense, increasing the risk of oxidative injury, particularly in the presence of dietary oxidants or concurrent copper deficiency [34,35]. These changes result in accelerated erythrocyte destruction, hemoglobin release into circulation, and hemoglobinuria, while sustained ATP reduction under chronic hypophosphatemia contributes to normocytic hypochromic anemia [25,26].

Postpartum hemoglobinuria (PPH) occurs sporadically in high producing dairy cows during the first weeks of lactation and is marked by sudden intravascular hemolysis with normocytic normochromic anemia (Figure 4 and hemoglobinuria (Figure 5) [16,36,37]. Severe hypophosphatemia is regarded as the primary metabolic trigger because it depletes ATP stores in erythrocytes and increases their fragility. Additional predisposing conditions include copper deficiency and hemolytic or oxidative constituents from Brassica species, beet residues, or fresh green forage [16,36]. Affected cows present with reduced milk output, pallor (Figure 6) or icterus, weakness, and dark red brown urine. Serum phosphorus concentrations are usually low, but may appear normal or elevated due to the release of intracellular phosphate during hemolysis [35]. Diagnosis requires the exclusion of alternative causes of hemolysis such as babesiosis, intoxications, or pronounced copper deficiency. Treatment outcomes vary. Blood transfusion may be required in severe anemia, whereas phosphorus therapy provides inconsistent improvement if oxidative damage predominates [33]. Preventive management targets adequate phosphorus and copper supply while avoiding forages and by-products rich in hemolytic or oxidizing plant constituents, particularly Brassica spp., beet tops, and rapidly fermentable green chop. Correcting trace mineral deficits and limiting dietary oxidant load reduces erythrocyte susceptibility to oxidative rupture and substantially lowers the risk of postpartum hemoglobinuria recurrence [16,33,36,37].

## 7. Phosphorus Requirements and Dietary Sources (e.g., per NRC)

Absorbability from different feeds (phytates; organic vs. inorganic forms). TMR formulation and Ca:P balance. Phytases and their role in improving P availability in monogastrics.

Phosphorus requirements in cattle depend on age, body weight, and level of production (growth, lactation, pregnancy). Nutritional standards (e.g., NRC, National Research Council) specify the minimum phosphorus concentration in dietary dry matter for different classes of animals. For dairy cows, the recommended dietary P concentration is approximately 0.30–0.40% of DM, with higher values applicable to very high-producing cows [20]. For example, a cow producing 40 kg milk/day should receive about 0.38–0.40% P, whereas a dry cow should receive ~0.25–0.30%. These guidelines translate into concrete amounts: a 650 kg cow consuming 20 kg DM/day should ingest roughly 60–80 g P/day (in lactation), while a growing heifer (400 kg) requires about 30–40 g P/day. Excess phosphorus supplementation is not beneficial. Numerous studies show that feeding above requirement does not further improve milk yield or fertility and merely increases fecal P excretion [38]. Only a small fraction of phosphorus supplied beyond requirement is retained (skeletal saturation may increase slightly); most is excreted—primarily in feces in ruminants (with absorbed surplus secreted in bile back into the gut) and partly in urine. From both economic and environmental perspectives, oversupplying P is wasteful and increases nutrient loading [39].

Dietary sources of phosphorus vary in content and bioavailability. Forages (grasses, silages) typically contain ~0.2–0.3% P (DM), part in available and part in organically bound forms. Cereal grains are richer (~0.3–0.4% P DM), but 60–80% of their phosphorus occurs as phytate, especially in brans and meals. Phytates are esters of myo-inositol and phosphoric acid that form complexes with proteins and minerals. In swine and poultry, phytate is poorly digested (lack of endogenous phytase), resulting in low P availability; phytate-bound phosphorus may be utilized at only 20–30% in monogastrics [40], hence the emphasis on available P in their formulations. In ruminants, the rumen microbiota efficiently hydrolyzes phytate, so P availability from cereal feeds is higher; apparent total-tract P digestibility in typical dairy rations often reaches ~60–70% [20].

Animal-derived feeds (milk, fish meal) provide phosphorus largely as organic phosphates readily cleaved by digestive enzymes and thus well-absorbed, but they are now rarely used in cattle diets (full milk for calves is an exception). Mineral sources include dicalcium phosphate (DCP), monocalcium phosphate (MCP), sodium phosphate, ammonium phosphate, and defluorinated phosphate rock. DCP typically contains ~18% P with high bioavailability (~80–95%). MCP contains ~22% P and is highly soluble in acidic conditions (favorable for ruminant gastric compartments), yielding excellent availability. Historically, bone meal (rich in Ca and P) was used on farms, but for safety reasons (BSE), its use in ruminants is prohibited [41].

Formulating total mixed rations (TMR) requires balancing phosphorus to established standards and paying close attention to the Ca:P ratio. In adult cows, a dietary calcium-to-phosphorus ratio of 1.5–2:1 is generally recommended (i.e., ~1.5–2 parts calcium per part phosphorus). An excessively low ratio (below 1:1, i.e., excess phosphorus relative to calcium) is undesirable as it can impair calcium absorption and promote secondary hyperparathyroidism with skeletal demineralization [20]. Excess dietary P binds calcium in the intestine as poorly soluble complexes and can stimulate PTH secretion (by lowering ionized Ca), thereby accelerating bone demineralization. Conversely, an excessively high Ca:P ratio (e.g., ≥3:1) may depress phosphorus absorption [20], as surplus calcium precipitates phosphate in the gut and reduces its solubility. Hence, Ca:P balance matters: not only the absolute P supply but also its proportion to calcium determines the effective utilization of both minerals. During the dry period, lower dietary calcium is often targeted to prevent milk fever; this does not obviate the need to meet a minimum P intake (e.g., 30–35 g P/day). It is important that calcium not excessively dominate over phosphorus, for example, on heavily limed pastures or with high-calcium buffers, since subclinical P deficiency has been observed under such conditions despite normal or elevated Ca, with adverse effects on appetite and fertility [42].

Improving P availability from feeds can be achieved with enzymatic additives. In monogastrics, microbial phytases are standard: added phytase hydrolyzes myo-inositol hexakisphosphate (phytate), releasing bound phosphorus and making it absorbable. Currently, >90% of poultry and ~70% of swine compound feeds include phytase to increase P utilization and reduce environmental excretion [43]. Phytase supplementation can raise plant P digestibility by several tens of percentage points, allowing a reduction in mineral phosphate inclusion, thereby lowering feed costs and decreasing phosphorus loading of soils and waters via manure. In ruminants, native rumen microbial phytase renders supplemental feed phytase generally unnecessary (possible exceptions include very high-producing cows on high-concentrate diets, and young ruminants with an underdeveloped rumen, where some benefit has been reported). In practice with broilers, for example, 500–1000 FYT/kg feed often permits a ~0.10% reduction in total dietary P without loss of growth performance, markedly reducing manure P [44].

Meeting cattle P requirements depends on choosing ingredients for both content and bioavailability, with mineral supplementation as needed, making strategic use of natural sources (good-quality corn silage ~0.25% P, young grasses ~0.30% P), and correcting deficits with validated mineral phosphates, while avoiding both deficiency (reduced performance, health issues) and excess (waste, health and environmental problems). Nutritionists should routinely analyze rations for P, especially when introducing new components (e.g., DDGS—high in P; beet pulp—low in P). In field practice, periodic blood phosphorus testing of a few representative cows is advisable, with dietary adjustments made when deviations are detected.

## 8. Disorders Associated with Excess Phosphorus: Excess Dietary P and Impaired Mineralization. Ca:P Ratio Imbalances—Risk of Uroliths and Secondary Hyperparathyroidism—Environmental Pollution (Phosphorus in Manure)

Although phosphorus deficiency is the more frequent problem in cattle nutrition, excessive dietary phosphorus can also cause disorders. An overly high P supply, particularly in the face of relative calcium deficiency, disrupts the balance of bone mineralization and endocrine control. As noted earlier, excess phosphorus leads to sustained stimulation of parathyroid hormone secretion (by lowering ionized Ca in blood and possibly via a direct effect of high Pi on the parathyroids), resulting in secondary hyperparathyroidism. Consequently, excess PTH drives the resorption of calcium (and phosphorus) from bone, producing demineralization despite abundant dietary P. This metabolic bone disease is termed nutritional osteodystrophy (especially in young, growing animals) and is characterized by proliferation of fibrous tissue in bone, skeletal tenderness, and sometimes deformities (e.g., mandibular/maxillary thickening—the “big head” phenotype in horses). In adult cattle, the effects of phosphorus excess are usually less dramatic, but long-term feeding of rations with an inverted Ca:P ratio (e.g., 1:2, twice as much P as Ca) can lead to fragile bones and osteoporosis [45]. In practice, however, cattle tolerate moderate deviations (up to 3:1) without major issues provided the total P intake meets requirements. According to [46], grazing cattle tolerate moderate Ca:P deviations (up to 3:1), provided the total intake of both minerals meets the physiological requirements.

One of the most common sequelae of excess dietary phosphorus is urolithiasis in cattle and other ruminants. High-P rations, particularly those with a large grain component or P-rich by-products such as DDGS, increase urinary phosphate excretion. When urine is alkaline (as is typical in herbivores) and there is insufficient luminal calcium to bind phosphorus in the gut, phosphate calculi can precipitate within the urinary tract. The characteristic stone in grain-fed feedlot cattle is struvite (magnesium ammonium phosphate, MgNH_4_PO_4_·6H_2_O), which forms in urine that is rich in phosphate and magnesium at a high pH. Clinical signs include dysuria, colic, and, with urethral obstruction, urinary retention and bladder rupture (“water belly”). The incidence of urolithiasis rises markedly in young feedlot animals fed high-grain diets with inverted Ca:P ratios (e.g., 1:1 or lower) [12]. Therefore, in intensive feeding systems, it is advisable to maintain an excess of calcium relative to phosphorus (approximately 2:1) to counteract phosphate precipitation in urine. Calcium binds excess phosphorus in the gastrointestinal tract and, to some extent, precipitates phosphates in the rumen. Prophylaxis of urolithiasis also includes urine acidification (e.g., ammonium chloride) and increasing water and salt intake, but the cornerstone is ensuring a dietary Ca:P ≥ 1.5:1. Notably, vitamin A deficiency (which compromises the urothelial lining) and low dietary fiber further predispose to stone formation. Practically, cases of urolithiasis in feedlot cattle should prompt verification of dietary phosphorus content, which is often excessive, particularly when rations contain high proportions of distillers grains or wheat bran [47].

Excess dietary phosphorus can also disrupt the metabolism of other minerals. As noted, high P intake depresses the utilization of magnesium and zinc. Formation of phytate–zinc complexes reduces Zn bioavailability; thus, high-grain diets (rich in phytate phosphorus) predispose cattle to subclinical zinc deficiency, manifested by poorer hoof and skin quality. Studies have shown that phosphate overfeeding can lower Zn concentrations in blood and liver despite adequate Zn intake. Conversely, very high phosphorus levels (or use of phosphate analogs, e.g., phosphate binders in humans) can reduce calcium absorption, precipitating tetany-like signs despite normal calcium intake [48]. In dairy cows, excess phosphorus during the dry period is a risk factor for postpartum milk fever. It can blunt adaptive mechanisms of calcium homeostasis, possibly by inhibiting vitamin D activation via FGF23. Historically, limiting dietary P in dry cows was recommended to prevent milk fever (analogous to restricting Ca); currently the emphasis is placed primarily on calcium, but very high P levels (>0.5% of DM) during the dry period are likewise avoided [23].

One must not overlook the environmental dimension of excess phosphorus. When feed contains more P than animals can utilize, the surplus is excreted in manure. Phosphorus in organic fertilizers (manure, slurry) is a significant source of surface-water contamination. Runoff of phosphates from fields, especially in soluble form or bound to soil particles, drives the eutrophication of water bodies, with excessive growth of algae and cyanobacteria and, consequently, oxygen depletion and die-offs of aquatic organisms [49]. This phenomenon, commonly referred to as algal bloom, is a major environmental problem in areas of intensive livestock production (e.g., regions with high densities of dairy or swine operations). Accordingly, many countries have enacted regulations that limit the maximum dietary P concentrations and restrict the land application of slurry during high-runoff periods.

From the farm’s perspective, excess phosphorus also represents an economic loss [50]. It is estimated that every 0.01% P fed above requirement in a cow’s ration equates to roughly 1.5–2.0 g of P wasted per day, or 0.5–0.7 kg of elemental P per cow per year. Given the price of feed phosphates, this constitutes avoidable cost [30]. Moreover, only 20–40% of the phosphorus entering a farm in feed is captured in products (milk, meat); the remainder represents potential environmental loss. It is therefore in the producer’s interest to precisely balance dietary P to meet, but not markedly exceed, the animal requirements [51].

Excess phosphorus can lead to: disordered bone mineralization (osteodystrophy secondary to hyperparathyroidism), urinary calculi (with low Ca:P), potentially poorer absorption of Zn, Mg, and Ca, and adverse environmental impacts (water eutrophication). The best strategy is to avoid overfeeding phosphorus by adhering to requirement-based guidelines, using phytase in monogastrics, and periodically analyzing feeds for P content. In the era of sustainable agriculture, minimizing phosphorus excretion has become a key objective of modern nutrition programs, reducing environmental pollution while maintaining high animal productivity.

## 9. Methods for Assessing Phosphorus Status

### 9.1. Measurement of P in Serum, Plasma, and Milk

Measurement of inorganic phosphorus in blood is the simplest and most commonly used method. Serum or plasma is analyzed for inorganic phosphate (Pi). In healthy adult cows, the reference range is approximately 4.5–6.5 mg/dL (1.45–2.10 mmol/L). Values below the reference range indicate hypophosphatemia; values above it indicate hyperphosphatemia. Blood phosphorus is often measured alongside calcium and magnesium (Mg) as part of a mineral profile [52]. It must be remembered, however, that blood P concentration varies with factors such as time of day, feed intake, insulin secretion, and physiological state (pregnancy, lactation). A single result should therefore be interpreted cautiously and in the context of other parameters and clinical findings. Despite these limitations, plasma Pi remains the primary tool for diagnosing phosphorus deficiency and is widely available in veterinary laboratories. The literature increasingly recommends monitoring phosphatemia in periparturient cows (e.g., 1 week prepartum and 1 week postpartum) to detect herd-level subclinical hypophosphatemia. For example, a mean P concentration < 4.0 mg/dL on day 4 postpartum suggests that the postpartum ration may be P-deficient or that adaptive mechanisms are failing to meet demand.

Milk phosphorus measurement. Bovine milk contains about 0.9 g P/L (primarily in casein and colloidal phosphates). In the setting of severe phosphorus deficiency, a decrease in milk P content has been observed; however, this parameter exhibits considerable individual variability. Studies indicate that milk P concentration can differ substantially among cows on similar diets (0.7–1.2 g/kg) [53]. Consequently, direct use of milk P as a mineral status indicator is challenging. Nonetheless, milk composition analysis (e.g., within dairy testing programs) for P could serve in the future as a rapid screening tool; ongoing work aims to develop models correlating milk P with a cow’s P balance [54]. For now, this measurement is primarily of research interest and is seldom used in field diagnostics.

### 9.2. Tissue Indicators (e.g., in Erythrocytes)

Since plasma phosphorus primarily reflects the extracellular pool and is subject to considerable variability, more stable intracellular indicators have been sought. One proposed parameter is the determination of phosphorus content in erythrocytes. Red blood cells contain substantial amounts of organic phosphates, such as 2,3-diphosphoglycerate (2,3-DPG) and adenosine triphosphate (ATP), which are essential for their proper function. Under conditions of phosphorus deficiency, the concentrations of 2,3-DPG and ATP in erythrocytes decrease, leading to increased susceptibility to hemolysis [37]. Studies have shown that cows with chronic phosphorus deficiency exhibit reduced erythrocyte phosphate and ATP content, as well as morphological alterations of red blood cells, including smaller size and irregular shape [55].

Measurement of intraerythrocytic inorganic phosphorus (P-RBC) or 2,3-DPG concentration could serve as a long-term indicator of phosphorus status, independent of transient fluctuations caused by, for example, insulin activity. However, these methods are labor-intensive and require specialized equipment, such as inductively coupled plasma optical emission spectrometry (ICP-OES) for P-RBC analysis.

Another potential tissue marker is bone phosphorus content, assessed, for example, through bone mineral density or percentage bone ash. In live animals, such measurements are difficult to perform non-invasively, although portable X-ray densitometry devices can be used to evaluate tail or rib bones. In field studies, rib biopsies have been applied to determine bone ash percentage, with low mineralization (<65% ash) indicating phosphorus deficiency.

Although tissue-based markers provide valuable scientific insight, clinical practice still relies predominantly on measurements performed in body fluids.

### 9.3. Monitoring of Metabolic Profiles (Postpartum Profile)

Many dairy herds are included in routine metabolic monitoring programs, particularly during the postpartum period. These profiles typically include measurements of glucose, β-hydroxybutyrate (BHB), non-esterified fatty acids (NEFA), urea, liver enzymes, as well as electrolytes and minerals (Ca, Mg, P). Regular (e.g., monthly) determination of phosphorus concentrations in selected groups of cows, dry cows, recently calved cows, and cows at peak lactation, allows for the detection of trends and potential nutritional issues. For example, if the median plasma P concentration in dry cows is 5.5 mg/dL, but in cows 10 days postpartum it drops to 3.5 mg/dL, this may indicate insufficient phosphorus supply during the transition period relative to lactational demands. Similarly, if high-yielding cows in peak lactation show plasma P concentrations below the reference range, it may suggest that the lactation diet is inadequate in phosphorus relative to milk production [56]. Such profile-based analysis enables proactive dietary adjustments before overt clinical signs of deficiency appear. In addition, concurrent blood testing allows for the evaluation of other related parameters, for instance, alkaline phosphatase (ALP) activity, which increases during intensive bone remodeling, a process that may accompany phosphorus deficiency. It is also advisable to monitor the serum Ca:P ratio, although its interpretation can be challenging since deviations may result from imbalances in one or both elements.

Best-practice metabolic monitoring includes serum Pi measurement during the close-up period and within the first 3–5 days postpartum. Additional biomarkers include alkaline phosphatase (reflecting bone mobilization), urinary phosphate-to-creatinine ratios, and erythrocyte ATP/2,3-DPG concentrations for research applications. Herd screening is recommended when >20–30% of cows exhibit Pi < 4.0 mg/dL postpartum.

### 9.4. Modern Methods: Molecular Biomarkers and Imaging

In recent years, research has focused on identifying new biomarkers and imaging techniques for assessing phosphorus status. An interesting candidate is the previously mentioned hormone fibroblast growth factor 23 (FGF23), which in humans shows a marked increase in serum concentration in cases of phosphorus excess and certain phosphate-related disorders. FGF23 is also detectable in bovine blood, although normative data are still lacking [57]. Theoretically, elevated FGF23 in cows could indicate phosphorus excess or renal insufficiency, whereas very low levels might reflect prolonged phosphorus deficiency [58].

Other investigated indicators include bone turnover markers, such as C-terminal telopeptide of type I collagen (CTX), a marker of bone resorption, as well as osteocalcin and bone-specific alkaline phosphatase, both markers of bone formation. Phosphorus deficiency stimulates bone resorption to release stored phosphorus, which should be reflected by elevated CTX levels in the blood [59]. Indeed, calves fed a low-phosphorus diet have been reported to show higher CTX and hydroxyproline concentrations (the latter also a resorption marker). However, results in cows are less consistent, multiple factors, including pregnancy and lactation, can obscure interpretation, and it remains unclear whether increased CTX reflects phosphorus deficiency or, for example, calcium mobilization around calving [60]. Nevertheless, ongoing studies suggest that in the future, mineral profile panels may be expanded to include such indicators.

Regarding imaging techniques, dual-energy X-ray absorptiometry (DEXA) is widely used in human medicine to assess bone density. While not routinely applied in livestock, it has been used experimentally to assess the impact of phosphorus deficiency on calf and cow bones. Results have confirmed that DEXA can detect mineral loss (e.g., reduced vertebral bone mineral density, BMD) associated with phosphorus-deficient diets [61]. In the future, portable devices such as bone ultrasonography or mobile X-ray units may enable rapid skeletal assessments in cows, providing indirect evidence of chronic phosphorus deficiency. At present, however, these approaches remain primarily within the research domain.

Currently, the assessment of phosphorus status in cattle relies mainly on the measurement of inorganic phosphorus in blood, alongside clinical observation and performance records. Feed analysis (phosphorus content) and production indicators (milk yield, reproductive performance) provide valuable information. In the future, more sensitive biomarkers and diagnostic tools may become available, enabling more precise management of phosphorus metabolism in dairy herds. Even now, it is clear that phosphorus monitoring should not be neglected, despite the frequent emphasis placed on calcium and energy balance.

### 9.5. Emerging Diagnostic Tools

Emerging approaches for phosphorus-status assessment include near-infrared spectroscopy of milk, metabolomic profiling of erythrocytes, serum FGF23 as a regulator-based biomarker, and imaging techniques such as high-resolution ultrasound and micro-CT for the early detection of skeletal demineralization. These tools may enable earlier identification of subclinical hypophosphatemia at the herd level.

## 10. Current Research Directions: Phosphorus as an Immunomodulatory Factor

An increasing number of studies focus on the impact of phosphorus status on the bovine immune system. Eisenberg et al. [18] demonstrated that maintaining cows in a state of moderate hypophosphatemia leads to reduced granulocyte counts and decreased phagocytic activity of neutrophils. Subsequent experiments are attempting to elucidate the molecular mechanisms underlying these observations. Key questions include whether phosphorus deficiency affects the expression of leukocyte surface receptors, disrupts the energy metabolism of lymphocytes (thereby inhibiting their proliferation), or modifies the cytokine profile secreted by immune cells.

The effects of phosphorus excess on immunity are also under investigation. For example, research is exploring whether high phosphate intake can induce chronic low-grade inflammation through monocyte activation by phosphate crystals, or whether it modulates innate immune responses.

These findings have practical implications: if phosphorus proves to be an important modulator of immunity, mineral nutrition strategies could be used to improve cattle health, for instance, by minimizing postpartum immunosuppression through the provision of optimal phosphorus supply [18,26,62,63].

### 10.1. Interactions with the Gut Microbiome

Another emerging area of interest is the effect of phosphorus on the composition and activity of the gastrointestinal microbiota. In the rumen, bacteria require phosphorus for nucleic acid synthesis, and their growth rate is partly limited by phosphorus availability. The recirculation of phosphates via saliva is a key mechanism supplying the ruminal microbiome with this element. Both in vitro and in vivo studies have shown that phosphorus deficiency in ruminal contents impairs fermentation: ammonia concentrations increase (reflecting reduced incorporation into microbial protein), pH rises, while volatile fatty acid (VFA) concentrations, particularly acetate, and microbial biomass production decrease [1]. Cellulolytic bacteria have proven especially sensitive to phosphorus deprivation. Fiber digestibility has been shown to decline under phosphorus deficiency, despite adequate dietary energy and nitrogen supply.

Ongoing research projects are examining changes in ruminal microbiome structure under varying dietary phosphorus levels. High-throughput 16S rRNA sequencing is used to identify which bacterial groups proliferate and which decline under phosphorus deficiency or excess. Preliminary results suggest that low phosphorus intake reduces the abundance of fiber-degrading *Fibrobacter* spp., while proteolytic bacteria may increase in relative abundance, which could explain the elevated ruminal ammonia levels observed [64,65].

There is also interest in determining whether excessive phosphorus supplementation may adversely affect microbial balance. Theoretically, this could promote the proliferation of less desirable taxa (e.g., Proteobacteria) or microorganisms associated with diarrhea in calves, due to increased phosphorus flow to the small and large intestines. A Korean study [66] found that diarrheic calves exhibited a significant increase in the relative abundance of Proteobacteria in fecal microbiota, along with a reduction in Bacteroidetes and Actinobacteria, suggesting that disturbances in water–electrolyte balance (e.g., excess phosphate reaching the intestine) may favor pathogenic strains [67].

Beyond the rumen, researchers are also examining the fecal microbiome and its relationship with phosphorus excretion. Manipulating the microbiota could potentially enhance phosphorus retention in the host and reduce environmental losses. This research direction aligns with the concept of “mineral prebiotics”, feed additives designed to promote beneficial microbial populations while improving mineral utilization, for example, by providing specific carbohydrates that enhance bacterial phosphorus uptake, with subsequent digestion of these bacteria in the small intestine [68,69,70].

### 10.2. Interactions with Copper, Zinc, and Vitamin D: Phosphorus Interplay with Other Macro- and Microelements

Phosphorus does not function in isolation within the body but interacts with various macro- and microelements. One of the most important relationships is its interplay with calcium and vitamin D. Calcium and phosphorus must be balanced for proper mineralization, while vitamin D regulates the absorption of both minerals. However, certain aspects remain unresolved, for example, the role of phosphorus in the development of milk fever. Why does excessive phosphorus intake during the dry period increase the incidence of postpartum hypocalcemia? Possible mechanisms include the inhibition of vitamin D activation via FGF23 or interference with intestinal calcium transport mechanisms in the prepartum period [23,30,71,72,73].

Another important interaction involves copper. Phosphorus and copper deficiencies often coexist, particularly in cattle grazing on peat or sandy soils. Both deficiencies can lead to anemia and reduced immune function. Cases of postpartum hemoglobinuria have been reported in which supplementation with both phosphorus and copper resulted in clinical improvement [35,74]. Researchers are investigating whether phosphorus affects copper absorption, as high dietary phosphorus can precipitate copper in the gastrointestinal tract, similar to iron or zinc, thereby reducing its bioavailability, or whether it influences ceruloplasmin levels, the primary copper-containing transport protein in plasma. Qureshi and Deeba [75] note that high phosphate concentrations can form insoluble complexes with divalent metal ions (including Ca, Fe, Zn) in the gastrointestinal tract, decreasing their bioavailability. A similar mechanism may apply to copper, though this requires further experimental verification. In the rumen, copper cations can bind with phosphate anions to form insoluble complexes, thereby reducing copper bioavailability and limiting its absorption in the lower gastrointestinal tract [76]. Reduced copper availability, for example, due to intestinal precipitation, lowers serum ceruloplasmin activity, which is considered a reliable indicator of copper status in cattle [77].

From an immunological perspective, the P–Cu–Zn interaction is particularly interesting. Copper and zinc are cofactors for antioxidant enzymes and transcription factors (e.g., NF-κB) that are essential for immune responses [78]. It is possible that phosphorus deficiency, by inducing metabolic stress, alters cellular requirements or the distribution of Cu and Zn. With regard to zinc, high-phosphorus diets, particularly those rich in phytates, may reduce zinc availability. Recent studies have examined whether phytase supplementation (which hydrolyzes phytates) can improve zinc status in feedlot cattle fed dried distillers grains with solubles (DDGS). Preliminary results are promising, showing that phytase not only increases the digestibility of phosphorus but also zinc and other trace elements, confirming strong inter-element interactions [79].

High phosphorus levels can also interfere with magnesium absorption [46], which is especially relevant in the context of grass tetany, where excess dietary potassium is the primary factor but high phosphorus combined with low magnesium may also predispose to tetanic symptoms. Current experiments using intestinal epithelial cell cultures (Caco-2) exposed to different combinations of P, Ca, and Mg aim to elucidate competitive or synergistic transport mechanisms across enterocytes. The outcomes of such studies could lead to improved mineral ratio recommendations in ruminant diets, potentially considering not only Ca:P but also Ca + Mg:P ratios [80].

### 10.3. The Role of FGF23 in Cattle and Its Physiological Significance

Fibroblast growth factor 23 (FGF23), first identified in humans as the principal phosphaturic hormone, has attracted considerable interest in the context of livestock physiology. The development of bovine-specific immunoassays for FGF23 has enabled investigations showing that its circulating concentration changes under different mineral statuses. In calves, diets with excess phosphorus increased plasma FGF23 concentrations, which was associated with reduced serum 1,25(OH)_2_D_3_ levels, analogous to observations in other species [81,82]. In dairy cows, FGF23 may be involved in adaptation to lactation. Around parturition, when plasma phosphorus declines, FGF23 concentrations are thought to decrease, facilitating maximal phosphorus retention through increased renal reabsorption and enhanced vitamin D activation. Several weeks later, when feed intake rises and phosphorus balance improves, FGF23 may increase to prevent excessive accumulation of phosphorus and calcium.

These hypotheses are currently being tested, with initial results indicating that FGF23 rises in cows fed high-phosphorus diets. For example, following phosphorus supplementation, increased FGF23 gene expression in bone tissue has been observed. Ma et al. [83] reported in a study of 116 multiparous Holstein cows that, from two weeks prepartum to 14 days postpartum, cows with subclinical hypocalcemia exhibited elevated prepartum phosphorus concentrations alongside reduced 1,25(OH)_2_D_3_ and FGF23 levels. This pattern suggests a physiological reduction in FGF23 immediately before lactation to maximize phosphorus retention. Similarly, Reinhardt et al. [4], in a review of mineral metabolism during the transition period in dairy cows, noted that in animal models (including rodents), elevated circulating phosphorus stimulates FGF23 synthesis in osteocytes, whereas low-phosphorus diets prevent FGF23-mediated suppression of 1α-hydroxylase, indirectly indicating reduced FGF23 under negative phosphorus balance before calving [84].

FGF23 not only regulates phosphate handling via the kidneys but also directly inhibits bone mineralization by increasing pyrophosphate production—a potent inhibitor of hydroxyapatite crystal formation, partly through stimulation of ectonucleotide pyrophosphatase/phosphodiesterase 1 (ENPP1) expression in osteocytes [85]. In response to high dietary phosphate, FGF23 induces pyrophosphate synthesis and reduces bone alkaline phosphatase activity, thereby weakening mineralization [86]. Whether FGF23 plays the same critical role in ruminants as in humans remains uncertain. Given their high salivary phosphorus recycling and other adaptive mechanisms, cattle may rely less on FGF23 for phosphate homeostasis.

Understanding the role of FGF23 in cattle has practical implications. For example, should phosphorus supplementation be tapered gradually to avoid abrupt FGF23 elevations that could disrupt calcium and vitamin D metabolism? Do bovine renal disorders, though relatively rare, lead to elevated FGF23, and if so, what are the consequences? Could FGF23 serve as a diagnostic marker of subclinical phosphorus overload before the onset of urolithiasis? Since FGF23 is linked to bone metabolism and inhibits mineralization via pyrophosphate production, excess dietary phosphorus could paradoxically impair skeletal mineral deposition in high-producing cows, notwithstanding that this mechanism has not been fully confirmed in ruminants.

### 10.4. Other Research Directions

Genetic research and selective breeding for improved phosphorus utilization efficiency are also advancing. Efforts are underway to identify candidate genes potentially linked to phosphorus utilization or more efficient phosphorus conservation within the body. If dairy lines with lower phosphorus requirements, for example, those able to extract more phosphorus from feed, could be selected, this could reduce feeding costs and the environmental footprint of milk production [87,88]. However, genetic differences in phosphorus metabolism are not yet well-characterized.

Another emerging area concerns the effect of phosphorus on soil microbiota and plants. For instance, intensive fertilization with manure rich in phosphorus can alter soil microbial populations and the availability of other minerals. In the context of sustainable agriculture, research is focusing on closing the phosphorus cycle: from feed, through the animal, manure, anaerobic digestion in biogas plants, and back to the soil and crops. Cattle play a pivotal role in this cycle, as their phosphorus utilization efficiency determines how much phosphorus must be imported into the system (e.g., as mineral fertilizer).

Aligned with this trend, studies are exploring feed additives that reduce phosphorus excretion, beyond phytase, examples include rumen-active yeasts that bind phosphorus or intestinal bacterial strains that increase phosphorus precipitation in manure, thereby reducing runoff into water bodies [89,90].

Research on phosphorus in cattle has evolved from fundamental physiology (e.g., elucidating hormonal mechanisms and its role in immunity) to practical applications (e.g., nutritional optimization and environmental phosphorus management). Increasing emphasis is being placed on aspects such as immunity and the microbiome, reflecting a broader shift in breeding goals toward not only high productivity but also improved animal health and minimized environmental impact. Phosphorus emerges as a unifying element linking these themes, explaining the strong scientific interest it continues to attract [91,92,93,94].

## 11. Recent Advances (2015–2025)

### 11.1. Endocrine Regulation in Transition Cows

During the transition from late gestation to early lactation, dairy cows activate hormonal axes to maintain Ca–P balance. FGF23, a bone-derived hormone (with co-receptor Klotho), is upregulated by high plasma phosphate and acts to lower blood P. It downregulates renal NaPi-2a/2c cotransporters (increasing P excretion) and inhibits renal 1α-hydroxylase, thereby suppressing active vitamin D (1,25(OH)_2_D_3_) synthesis [95]. In contrast, PTH is secreted by the parathyroid in response to low blood Ca. PTH stimulates osteoclastic bone resorption (releasing Ca and P), increases renal Ca reabsorption and urinary P excretion, and stimulates 1α-hydroxylase to raise 1,25(OH)_2_D_3_ [96]. Active vitamin D promotes intestinal Ca and P absorption, creating negative feedback on PTH. In effect, PTH and vitamin D act in concert to elevate Ca (at the cost of P loss), whereas FGF23 opposes PTH’s effect on vitamin D to prevent phosphate overload [85,97]

In transition cows, these pathways are highly dynamic. At calving, the abrupt drop in blood Ca triggers a marked PTH surge, mobilizing Ca from bone and upregulating 1,25(OH)_2_D_3_ synthesis [98]. Recent studies show that dietary phosphorus restriction (for example via zeolite binders) lowers plasma P, which in turn suppresses FGF23 production. This relieves FGF23’s inhibition of 1α-hydroxylase, allowing higher 1,25(OH)_2_D_3_ and improved Ca absorption [97] Indeed, cows fed a P-binding supplement had significantly lower blood P and the highest plasma Ca pre- and postpartum [95]. Correspondingly, very low-P diets yield lower PTH and higher 1,25(OH)_2_D_3_ concentrations. Together, these findings suggest that in P-restricted cows, FGF23 falls (enhancing vitamin D activity) and PTH requirement is reduced, thereby bolstering calcium status around calving [97].

### 11.2. Subclinical Hypophosphatemia: Prevalence and Impact

Subclinical hypophosphatemia (SHP) is typically defined as plasma inorganic P below ≈1.3 mmol/L (~4 mg/dL) [96], a level at which leukocyte function is known to decline. Recent surveys report SHP is common in fresh cows. For example, one large French study (371 cows on commercial farms) found 37% of cows had SHP within 24–48 h postpartum; other studies reported a 15–55% prevalence [96]. In a controlled cohort on a balanced diet, the incidence was lower (≈17%) [98]. These data indicate SHP is often present even when overt disease is absent.

Risk factors for SHP mirror those for hypocalcemia. Higher parity markedly increases SHP risk (odds ratios ≈ 2.3–2.9 for 3rd–4th lactations) and cows with concurrent subclinical hypocalcemia were ~3 times more likely to become hypophosphatemic [96]. Cows fed moderately high pre-partum dietary P had reduced SHP risk (possibly via greater reserve), whereas anionic diets showed only a modest protective trend. In one study, multiparous cows on a standard diet had 43% SHP vs. 33% on a slightly acidified diet [96]. Overall, inadequate dry-cow P intake, high lactational demand, and low feed intake (e.g., illness) predispose to SHP.

The clinical significance of SHP remains under investigation. Although severe hypophosphatemia can cause postparturient hemoglobinuria and impaired appetite in individual cows, the effects of moderate SHP are subtle. Notably, plasma [P] is a poor marker of total P status: significant blood P drops do not reliably reflect intracellular P pools or chronic P deficiency [99]. However, severe P deficiency has been linked to immune dysfunction. The threshold of 1.3 mmol/L is based on work showing leukocyte impairment below this level [96]. Subclinical P deficiency may also worsen energy balance or mobility by contributing to muscle weakness and bone remodeling, although firm data in well-fed herds are limited. In practice, SHP is often detected only by blood chemistry panels (e.g., monitoring all fresh cows), and its importance is assessed case by case.

### 11.3. Models of Phosphorus Homeostasis

Compartmental metabolic models: Researchers have constructed mechanistic models to simulate P flux. For instance, [100] built a dynamic model (in a continuous simulation language) tracking inorganic, organic, and phytate P through the rumen, intestines, saliva, blood, and milk. This model predicted that roughly 65% of absorbed P is recycled to the rumen via saliva, about 30% goes to milk, and only ~1% is lost in urine. It highlighted bone formation/resorption and intestinal absorption as primary control points for plasma P (since bone P balance depended on dietary P and ruminal phytase) [100].

Nutritional requirement models: Systems like CNCPS or INRA’s models compute P balance based on intake, milk output, growth and excretion fractions. These aggregate models assume fixed P use for milk and maintenance; they do not explicitly simulate rapid endocrine feedback loops.

Dynamic endocrine models: Modern understanding calls for models that include hormone kinetics. For example, glucose infusion experiments show a rapid insulin-driven intracellular P shift (plasma P falls ~35% in 1 h post-dextrose) [99], a process not captured by static feeding models. Recent work [101] also points to changes in tissue hydration and cellular pools: postpartum cows lose liver water and total liver P content (though cytosolic [P] stays constant). In future, integrated models could incorporate such factors, endocrine triggers (PTH, FGF23, insulin), saliva recycling, bone reserves, and intracellular pools, to better predict P dynamics beyond the classic gut–bone–kidney paradigm [100].

## 12. Conclusions

Phosphorus plays a central regulatory role in energy metabolism, neuromuscular function, bone turnover, and immune competence in dairy cattle. Periparturient cows are at the highest risk of negative phosphorus balance due to the sudden transfer of P into colostrum and milk and reduced dry matter intake. Transient decreases in plasma Pi are part of normal metabolic adaptation, yet prolonged or severe hypophosphatemia contributes to postpartum recumbency, impaired uterine clearance, reduced feed intake, and increased susceptibility to infectious complications.

Subclinical hypophosphatemia remains underdiagnosed and can compromise early lactation performance, delay recovery from metabolic stress, and predispose to secondary inflammatory and reproductive disorders. Accurate monitoring of Pi alongside Ca, Mg, and endocrine regulators (PTH, FGF23, calcitriol) is required for proper interpretation of mineral status in fresh cows.

Phosphorus supplementation should neither be uniform nor excessive. Instead, it should be tailored to lactation stage, dietary Ca:P ratio, rumen buffering conditions, and documented postpartum P decline. Balanced P nutrition reduces the incidence of recumbency in the absence of hypocalcemia, improves immune function, and prevents long-term skeletal depletion.

Continued research on molecular biomarkers (e.g., FGF23), erythrocyte ATP dynamics, and early detection tools will advance precision mineral management in dairy herds. Sustainable phosphorus strategies must integrate productive performance with environmental stewardship, minimizing P excretion while maintaining optimal cow health.

## 13. Practical Recommendations

Effective management of phosphorus status in dairy cattle requires targeted and stage-specific intervention rather than uniform supplementation. Postpartum metabolic monitoring should routinely include inorganic phosphate alongside calcium and magnesium, as the period between the third and tenth day after calving represents the highest risk for clinically relevant declines. Phosphorus intake must be adjusted to physiological state: cows in early lactation require clearly increased supply, whereas dry cows and heifers should not receive excessive supplementation. Maintaining an appropriate dietary Ca:P ratio, generally within the range of 1.5–2:1, remains essential for efficient mineral absorption and endocrine stability. Nutritional decisions should follow requirement-based models, such as NRC 2021 [46], avoiding oversupply that fails to improve performance yet markedly elevates fecal phosphorus losses and environmental burden. Feeding strategies that alter rumen acidity or increase urinary phosphate excretion, including high-concentrate rations or anionic diets prepartum, warrant parallel reassessment of mineral balance. In practice, phosphorus management should integrate productivity goals with environmental stewardship, reducing excretion-derived eutrophication risks while safeguarding immune competence, neuromuscular function, and skeletal integrity. Particular attention is advised for fresh cows, high-yielding multiparous animals, and individuals recovering from metabolic disorders, as rapid correction of persistent postpartum hypophosphatemia supports early lactation performance, accelerates recovery from metabolic stress, and improves overall herd resilience.

## Figures and Tables

**Figure 1 animals-16-00130-f001:**
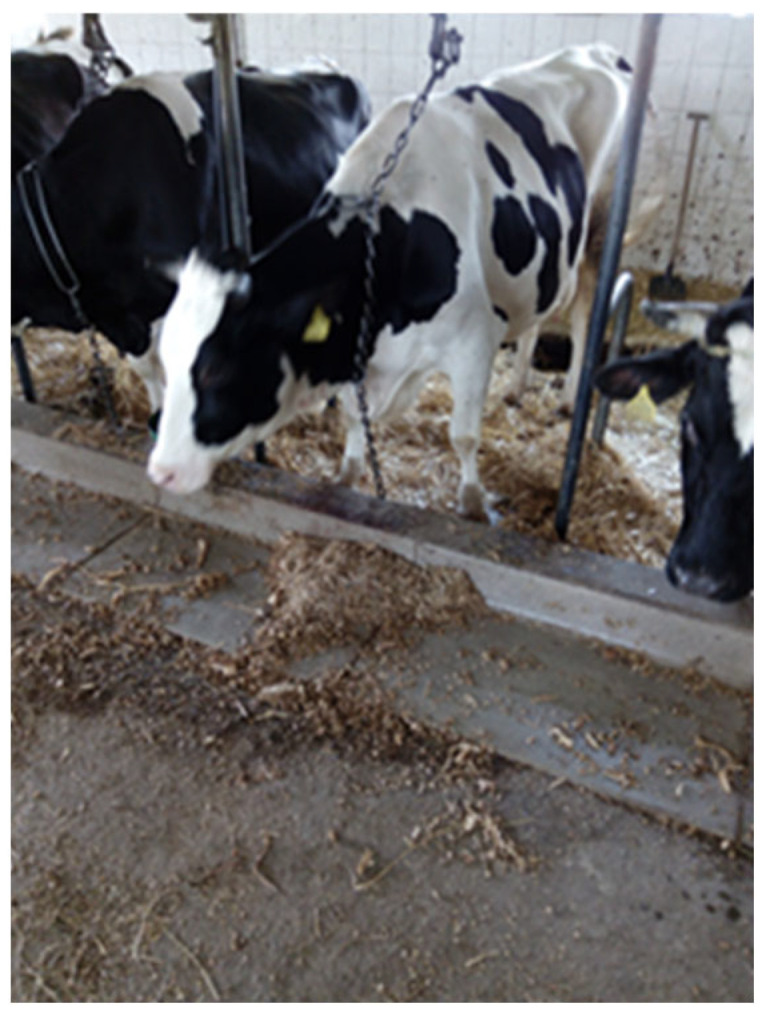
Loss of appetite in cow.

**Figure 2 animals-16-00130-f002:**
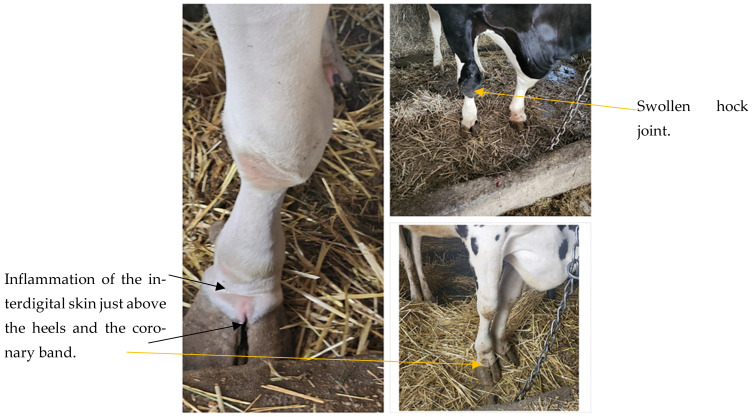
Swollen hock joint, inflammation of the interdigital skin just above the heels and the coronary band.

**Figure 3 animals-16-00130-f003:**
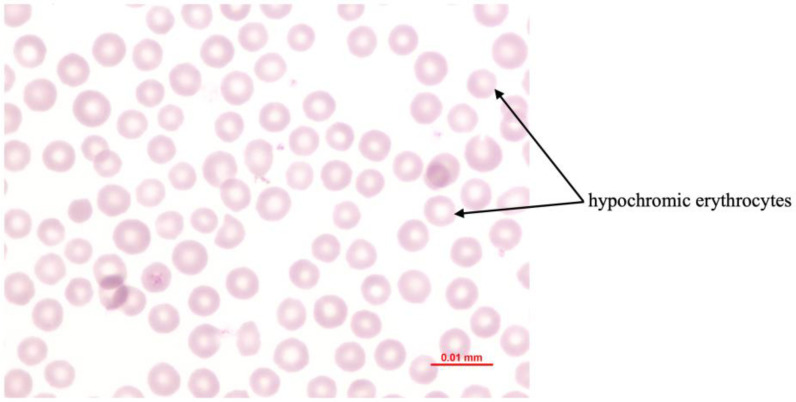
Normocytic hypochromic anemia.

**Figure 4 animals-16-00130-f004:**
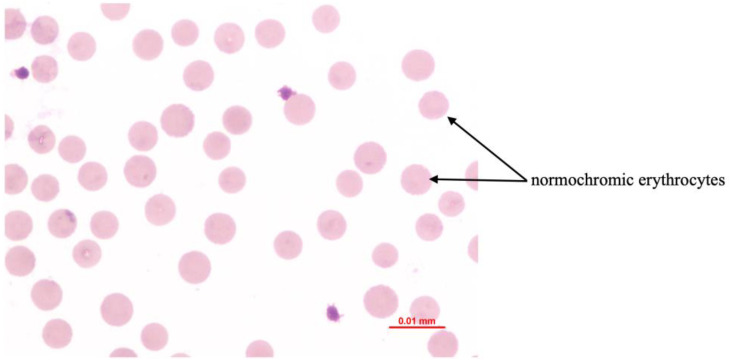
Normocytic, normochromic anemia.

**Figure 5 animals-16-00130-f005:**
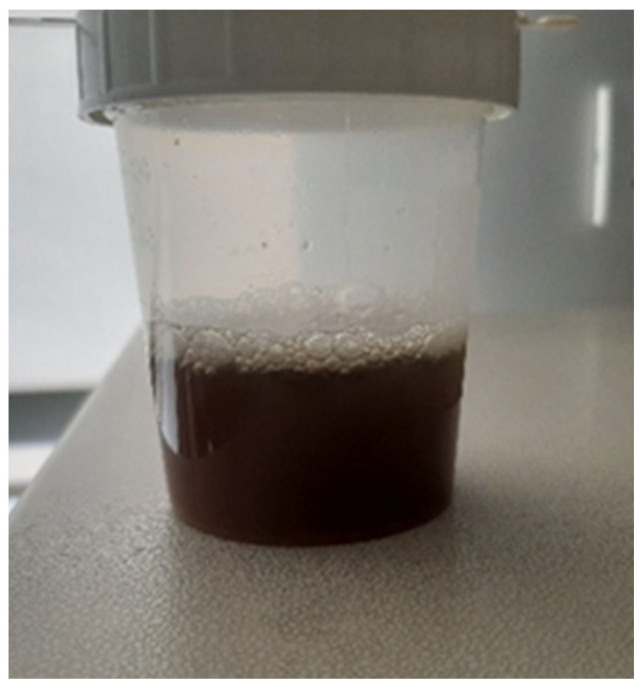
Hemoglobinuria.

**Figure 6 animals-16-00130-f006:**
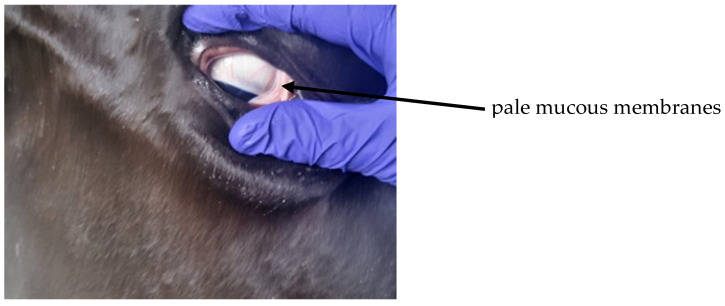
Pale mucous membranes.

**Table 1 animals-16-00130-t001:** Effects of Phosphorus Deficiency on Rumen Fermentation.

Parameter	Change in P Deficiency
Rumen NH_3_–N	↑ (increased)
Rumen pH	↑
Total VFA (short-chain fatty acids)	↓ (decreased)
Microbial protein yield	↓
Cellulose/Hemicellulose digestion	↓ (decreased)
Starch digestion	↔ (unchanged)

## Data Availability

All data are provided in the text.

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
