# Peer review of "Phosphorus Metabolism and Function in Ruminants: Current Knowledge"

_animals, 2026, doi:10.3390/ani16010130_

Round 1

Reviewer 1 Report

Comments and Suggestions for Authors
  1. The structure of the section "2. Physiology of phosphorus" is poorly organized and contains misplaced content, requiring reorganization;

  2. The annotations in the figures throughout the paper are insufficiently clear, and some figures lack control groups;

  3. The review content of the entire paper is somewhat superficial and fails to adequately reflect current research progress.

Comments on the Quality of English Language

The English expression in this paper is relatively adequate but lacks clarity and precision, necessitating further revision.

Author Response

 "Please see the attachment." in the box if you only upload an attachment.

Reviewer 2 Report

Comments and Suggestions for Authors

The manuscript provides a comprehensive review on the effect of phosphorus deficiency on acute-phase proteins in cows during the postpartum period. It highlights the critical roles phosphorus plays in various physiological functions, particularly in immune response, muscle function, and bone health. The article is well-organized, the writing is clear, and the authors do an excellent job synthesizing existing knowledge. However, there are several areas where the manuscript could be strengthened to improve clarity and impact.

  1. The introduction clearly defines phosphorus's importance. However, it would benefit from a brief mention of the broader implications for dairy farming productivity and animal welfare in areas where phosphorus deficiency is common.
  2. While the review covers a broad range of studies, it would be useful to more explicitly highlight recent research advancements or conflicting findings. The integration of such sources could help show the development of the field.
  3. Immunological Effects: The section on the immune system could delve deeper into the mechanisms through which phosphorus affects immune cell proliferation and function. Additionally, it would benefit from more discussion on the long-term consequences of subclinical hypophosphatemia.
  4. Immunological Effects: The section on the immune system could delve deeper into the mechanisms through which phosphorus affects immune cell proliferation and function. Additionally, it would benefit from more discussion on the long-term consequences of subclinical hypophosphatemia.
  5. Phosphorus Deficiency and Musculoskeletal Health: The connection between phosphorus deficiency and musculoskeletal health is well covered. However, providing more detailed references to studies on bone health and chronic effects of phosphorus depletion would enhance this section.
  6. Phosphorus and Reproductive Performance: The manuscript mentions reduced reproductive efficiency but could benefit from a deeper exploration of how phosphorus deficiency directly impacts hormonal balance and reproductive outcomes.
  7. Metabolic Monitoring: The section on phosphorus status assessment through metabolic monitoring could be expanded to include more details on best practices for dairy farmers and veterinarians, as well as specific biomarkers for early detection.
  8. Emerging Research: It would be beneficial to add a discussion on emerging techniques, such as molecular biomarkers or imaging techniques, to monitor phosphorus status more effectively.

Author Response

(The authors gave the same response as above.)

Reviewer 3 Report

Comments and Suggestions for Authors

Comments and Suggestions for Authors

Title: The effect of phosphorus deficiency on acute phase proteins in cows during the postpartum period

This manuscript provides a comprehensive review of phosphorus deficiency in dairy cows, with a particular focus on its impact during the postpartum period. The authors effectively synthesise current knowledge on phosphorus metabolism, hormonal regulation, and the clinical consequences of deficiency, including its effects on acute-phase proteins, erythrocyte integrity, and immune function.

Overall, the review is scientifically sound and relevant to veterinary researchers and animal nutrition literature.

However, the title suggests a focus on acute-phase proteins, but this aspect is underdeveloped compared to other physiological systems. A deeper analysis of specific proteins (haptoglobin and serum amyloid A) and their diagnostic relevance would strengthen the manuscript.

While the review is detailed and informative, it could benefit from clearer emphasis on recent discoveries or emerging research directions, such as FGF23 as a biomarker.

The manuscript presentation is overly dense, which significantly hinders readability and comprehension. The current structure would benefit from clearer segmentation, with more concise sections and the inclusion of summary tables and diagrams to highlight key information.

Specific Textual Suggestions:

Lines 38-39: The phrase "emerging links between phosphorus balance, immune competence, and environmental sustainability" is unclear; consider briefly elaborating or clarifying what these links are.

Lines 644-647: The sentence “Vitamin and mineral nutrition of grazing cattle.; [51]…” seems fragmented. Consider revising for clarity: “According to [51], grazing cattle tolerate moderate Ca:P deviations (up to 3:1) without major issues, provided total intake meets requirements”. 

Lines 1009-1010: “Phosphorus should neither be oversupplied nor given uniformly…”, consider rephrasing for clarity: Phosphorus supplementation should be tailored, avoiding both excess and uniform dosing.

Additionally, the conclusions and practical recommendations should be more succinctly formulated to enhance their clarity and applicability. A streamlined format will enhance the manuscript’s impact, making it more accessible and useful to both researchers and practitioners.

Please revise the overall structure of the manuscript, reformulating the content in a concise and intelligible manner to enable quick and complete extraction of essential information.

A clearer organisation will improve the manuscript’s accessibility, applicability, and effectiveness in technical use.

Recommendation: Major revision

Author Response

(The authors gave the same response as above.)

Round 2

Reviewer 1 Report

Comments and Suggestions for Authors

none

Comments on the Quality of English Language

The English expression in this paper is relatively adequate but lacks clarity and precision, necessitating further revision.

Reviewer 3 Report

Comments and Suggestions for Authors

Dear Authors,

Thank you for your answers. I have no further questions.